# TRAINING INDIVIDUALLY FAIR ML MODELS WITH SENSITIVE SUBSPACE ROBUSTNESS

**Mikhail Yurochkin**
IBM Research
MIT-IBM Watson AI Lab
`mikhail.yurochkin@ibm.com`

**Amanda Bower**[†]**, Yuekai Sun**[‡]
Department of Mathematics[†]
Department of Statistics[‡]
University of Michigan
`{amandarg,yuekai}@umich.edu`

## ABSTRACT

We consider training machine learning models that are fair in the sense that their performance is invariant under certain sensitive perturbations to the inputs. For example, the performance of a resume screening system should be invariant under changes to the gender and/or ethnicity of the applicant. We formalize this notion of algorithmic fairness as a variant of individual fairness and develop a distributionally robust optimization approach to enforce it during training. We also demonstrate the effectiveness of the approach on two ML tasks that are susceptible to gender and racial biases.

## 1 INTRODUCTION

Machine learning (ML) models are gradually replacing humans in high-stakes decision making roles. For example, in Philadelphia, an ML model classifies probationers as high or low-risk (Metz & Satariano, 2020). In North Carolina, "analytics" is used to report suspicious activity and fraud by Medicaid patients and providers (Metz & Satariano, 2020). Although ML models appear to eliminate the biases of a human decision maker, they may perpetuate or even exacerbate biases in the training data (Barocas & Selbst, 2016). Such biases are especially objectionable when it adversely affects underprivileged groups of users (Barocas & Selbst, 2016).

In response, the scientific community has proposed many mathematical definitions of algorithmic fairness and approaches to ensure ML models satisfy the definitions. Unfortunately, this abundance of definitions, many of which are incompatible (Kleinberg et al., 2016; Chouldechova, 2017), has hindered the adoption of this work by practitioners. There are two types of formal definitions of algorithmic fairness: group fairness and individual fairness. Most recent work on algorithmic fairness considers group fairness because it is more amenable to statistical analysis (Ritov et al., 2017). Despite their prevalence, group notions of algorithmic fairness suffer from certain shortcomings. One of the most troubling is there are many scenarios in which an algorithm satisfies group fairness, but its output is blatantly unfair from the point of view of individual users (Dwork et al., 2011).

In this paper, we consider individual fairness instead of group fairness. Intuitively, an individually fair ML model treats similar users similarly. Formally, an ML model is a map $h : \mathcal{X} \to \mathcal{Y}$, where $\mathcal{X}$ and $\mathcal{Y}$ are the input and output spaces. The leading notion of individual fairness is metric fairness (Dwork et al., 2011); it requires

$$d_y(h(x_1), h(x_2)) \leq L d_x(x_1, x_2) \text{ for all } x_1, x_2 \in \mathcal{X}, \tag{1.1}$$

where $d_x$ and $d_y$ are metrics on the input and output spaces and $L \geq 0$ is a Lipschitz constant. The fair metric $d_x$ encodes our intuition of which samples should be treated similarly by the ML model. We emphasize that $d_x(x_1, x_2)$ being small does not imply $x_1$ and $x_2$ are similar in all respects. Even if $d_x(x_1, x_2)$ is small, $x_1$ and $x_2$ may differ in certain problematic ways, e.g. in their protected/sensitive attributes. This is why we refer to pairs of samples $x_1$ and $x_2$ such that $d_x(x_1, x_2)$ is small as *comparable* instead of similar.

Despite its benefits, individual fairness was dismissed as impractical because there is no widely accepted fair metric for many ML tasks. Fortunately, there is a line of recent work on learning the

fair metric from data (Ilvento, 2019; Wang et al., 2019). In this paper, we consider two data-driven choices of the fair metric: one for problems in which the sensitive attribute is reliably observed, and another for problems in which the sensitive attribute is unobserved (see Appendix B).

The rest of this paper is organized as follows. In Section 2, we cast individual fairness as a form of robustness: robustness to certain sensitive perturbations to the inputs of an ML model. This allows us to leverage recent advances in adversarial ML to train individually fair ML models. More concretely, we develop an approach to audit ML models for violations of individual fairness that is similar to adversarial attacks (Goodfellow et al., 2014) and an approach to train ML models that passes such audits (akin to adversarial training (Madry et al., 2017)). We justify the approach theoretically (see Section 3) and empirically (see Section 4).

## 2 FAIRNESS THROUGH (DISTRIBUTIONAL) ROBUSTNESS

To motivate our approach, imagine an auditor investigating an ML model for unfairness. The auditor collects a set of audit data and compares the output of the ML model on comparable samples in the audit data. For example, to investigate whether a resume screening system is fair, the auditor may collect a stack of resumes and change the names on the resumes of Caucasian applicants to names more common among the African-American population. If the system performs worse on the edited resumes, then the auditor may conclude the model treats African-American applicants unfairly. Such investigations are known as **correspondence studies**, and a prominent example is Bertrand & Mullainathan's celebrated investigation of racial discrimination in the labor market. In a correspondence study, the investigator looks for inputs that are comparable to the training examples (the edited resumes in the resume screening example) on which the ML model performs poorly. In the rest of this section, we formulate an optimization problem to find such inputs.

### 2.1 FAIR WASSERSTEIN DISTANCES

Recall $\mathcal{X}$ and $\mathcal{Y}$ are the spaces of inputs and outputs. To keep things simple, we assume that the ML task at hand is a classification task, so $\mathcal{Y}$ is discrete. We also assume that we have a fair metric $d_x$ of the form

$$d_x(x_1, x_2)^2 \triangleq \langle x_1 - x_2, \Sigma(x_1 - x_2) \rangle^{\frac{1}{2}},$$

where $\Sigma \in \mathbf{S}_+^{d \times d}$. For example, suppose we are given a set of $K$ "sensitive" directions that we wish the metric to ignore; *i.e.* $d(x_1, x_2) \ll 1$ for any $x_1$ and $x_2$ such that $x_1 - x_2$ falls in the span of the sensitive directions. These directions may be provided by a domain expert or learned from data (see Section 4 and Appendix B). In this case, we may choose $\Sigma$ as the orthogonal complement projector of the span of the sensitive directions. We equip $\mathcal{X}$ with the fair metric and $\mathcal{Z} \triangleq \mathcal{X} \times \mathcal{Y}$ with

$$d_z((x_1, y_1), (x_2, y_2)) \triangleq d_x(x_1, x_2) + \infty \cdot \mathbf{1}\{y_1 \neq y_2\}.$$

We consider $d_z^2$ as a transport cost function on $\mathcal{Z}$. This cost function encodes our intuition of which samples are comparable for the ML task at hand. We equip the space of probability distributions on $\mathcal{Z}$ with the fair Wasserstein distance

$$W(P, Q) = \inf_{\Pi \in \mathcal{C}(P,Q)} \int_{\mathcal{Z} \times \mathcal{Z}} c(z_1, z_2) d\Pi(z_1, z_2),$$

where $\mathcal{C}(P, Q)$ is the set of couplings between $P$ and $Q$. The fair Wasserstein distance inherits our intuition of which samples are comparable through the cost function; *i.e.* the fair Wasserstein distance between two probability distributions is small if they are supported on comparable areas of the sample space.

### 2.2 AUDITING ML MODELS FOR ALGORITHMIC BIAS

To investigate whether an ML model performs disparately on comparable samples, the auditor collects a set of audit data $\{(x_i, y_i)\}_{i=1}^n$ and solves the optimization problem

$$\max_{P: W(P, P_n) \leq \epsilon} \int_{\mathcal{Z}} \ell(z, h) dP(z), \tag{2.1}$$

where $\ell : \mathcal{Z} \times \mathcal{H} \to \mathbf{R}_+$ is a loss function, $h$ is the ML model, $P_n$ is the empirical distribution of the audit data, and $\epsilon > 0$ is a small tolerance parameter. We interpret $\epsilon$ as a moving budget that the auditor may expend to discover discrepancies in the performance of the ML model. This budget forces

the auditor to avoid moving samples to incomparable areas of the sample space. We emphasize that equation 2.1 detects *aggregate* violations of individual fairness. In other words, although the violations that the auditor's problem detects are individual in nature, the auditor's problem is only able to detect aggregate violations. We summarize the implicit notion of fairness in equation 2.1 in a definition.

**Definition 2.1** (distributionally robustly fair (DRF)). *An ML model $h : \mathcal{X} \to \mathcal{Y}$ is $(\epsilon, \delta)$-distributionally robustly fair (DRF) WRT the fair metric $d_x$ iff*

$$\max_{P:W(P,P_n)\leq\epsilon} \int_{\mathcal{Z}} \ell(z,h)dP(z) \leq \delta. \tag{2.2}$$

Although equation 2.1 is an infinite-dimensional optimization problem, it is possible to solve it exactly by appealing to duality. Blanchet & Murthy showed that the dual of equation 2.1 is

$$\sup_{P:W(P,P_n)\leq\epsilon} \mathbb{E}_P\big[\ell(Z,h)\big] = \inf_{\lambda\geq 0}\{\lambda\epsilon + \mathbb{E}_{P_n}\big[\ell^c_\lambda(Z,h)\big]\},$$
$$\ell^c_\lambda((x_i,y_i),h) \triangleq \sup_{x\in\mathcal{X}} \ell((x,y_i),\theta) - \lambda d_x(x,x_i). \tag{2.3}$$

This is a univariate optimization problem, and it is amenable to stochastic optimization. We describe a stochastic approximation algorithm for equation 2.3 in Algorithm 1. Inspecting the algorithm, we see that it is similar to the PGD algorithm for adversarial attack.

---

**Algorithm 1** stochastic gradient method for equation 2.3

---

**Require:** starting point $\hat{\lambda}_1$, step sizes $\alpha_t > 0$
1: **repeat**
2:      draw mini-batch $(x_{t_1}, y_{t_1}), \ldots, (x_{t_B}, y_{t_B}) \sim P_n$
3:      $x^*_{t_b} \leftarrow \arg\max_{x\in\mathcal{X}} \ell((x,y_{t_b}),h) - \lambda d_x(x_{t_b},x), b \in [B]$
4:      $\hat{\lambda}_{t+1} \leftarrow \max\{0, \hat{\lambda}_t - \alpha_t(\epsilon - \frac{1}{B}\sum_{b=1}^{B} d_x(x_{t_b}, x^*_{t_b}))\}$
5: **until** converged

---

It is known that the optimal point of equation 2.1 is the discrete measure $\frac{1}{n}\sum_{i=1}^{n} \delta_{(T_\lambda(x_i),y_i)}$, where $T_\lambda : \mathcal{X} \to \mathcal{X}$ is the *unfair map*

$$T_\lambda(x_i) \leftarrow \arg\max_{x\in\mathcal{X}} \ell((x,y_i),h) - \lambda d_x^2(x,x_i). \tag{2.4}$$

We call $T_\lambda$ an unfair map because it reveals unfairness in the ML model by mapping samples in the audit data to comparable areas of the sample space that the system performs poorly on. We note that $T_\lambda$ may map samples in the audit data to areas of the sample space that are not represented in the audit data, thereby revealing disparate treatment in the ML model not visible in the audit data alone. We emphasize that $T_\lambda$ more than reveals disparate treatment in the ML model; it *localizes* the unfairness to certain areas of the sample space.

We present a simple example to illustrating fairness through robustness (a similar example appeared in Hashimoto et al. (2018)). Consider the binary classification dataset shown in Figure 1. There are two subgroups of observations in this dataset, and (sub)group membership is the protected attribute (*e.g.* the smaller group contains observations from a minority subgroup). In Figure 1a we see the decision heatmap of a vanilla logistic regression, which performs poorly on the blue minority subgroup. The two subgroups are separated in the horizontal direction, so the horizontal direction is the sensitive direction. Figure 1b shows that such classifier is unfair with respect to the corresponding fair metric, i.e. the *unfair map* equation 2.4 leads to significant loss increase by transporting mass along the horizontal direction with very minor change of the vertical coordinate.

**Comparison with metric fairness**    Before moving on to training individually fair ML models, we compare DRF with metric fairness equation 1.1. Although we concentrate on the differences between the two definitions here, they are more similar than different: both formalize the intuition that the outputs of a fair ML model should perform similarly on comparable inputs. That said, there are two main differences between the two definitions. First, instead of requiring the output of the ML model to be similar on all inputs comparable to a training example, we require the output to be similar to the training label. Thus DRF not only enforces similarity of the output on comparable inputs, but also accuracy of the ML model on the training data. Second, DRF considers differences

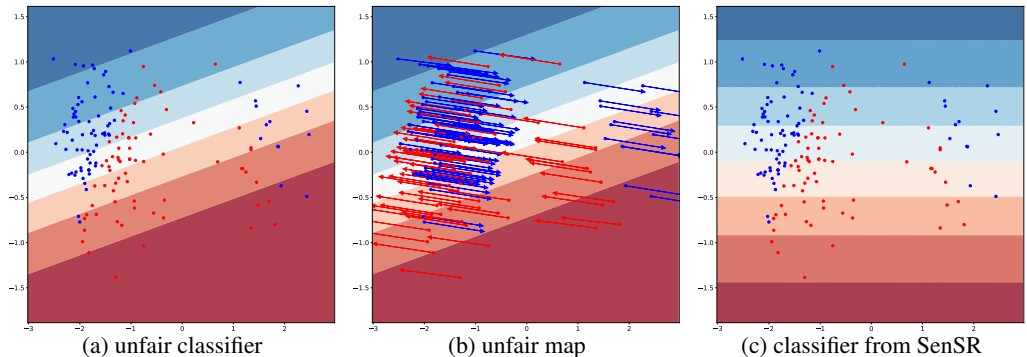

(a) unfair classifier $\qquad$ (b) unfair map $\qquad$ (c) classifier from SenSR

Figure 1: Figure (a) depicts a binary classification dataset in which the minority group shown on the right of the plot is underrepresented. This tilts the logistic regression decision boundary in favor of the majority group on the left. Figure (b) shows the unfair map of the logistic regression decision boundary. It maps samples in the minority group towards the majority group. Figure (c) shows an algorithmically fair classifier that treats the majority and minority groups identically.

between datasets instead of samples by replacing the fair metric on inputs with the fair Wasserstein distance induced by the fair metric. The main benefits of this modifications are (i) it is possible to optimize equation 2.1 efficiently, (ii) we can show this modified notion of individual fairness generalizes.

## 2.3 FAIR TRAINING WITH SENSITIVE SUBSPACE ROBUSTNESS

We cast the fair training problem as training supervised learning systems that are robust to sensitive perturbations. We propose solving the minimax problem

$$\inf_{h \in \mathcal{H}} \sup_{P:W(P,P_n) \leq \epsilon} \mathbb{E}_P\big[\ell(Z,h)\big] = \inf_{h \in \mathcal{H}} \inf_{\lambda \geq 0} \lambda\epsilon + \mathbb{E}_{P_n}\big[\ell_\lambda^c(Z,h)\big], \qquad (2.5)$$

where $\ell_\lambda^c$ is defined in equation 2.3. This is an instance of a distributionally robust optimization (DRO) problem, and it inherits some of the statistical properties of DRO. To see why equation 2.5 encourages individual fairness, recall the loss function is a measure of the performance of the ML model. By assessing the performance of an ML model by its worse-case performance on hypothetical populations of users with perturbed sensitive attributes, minimizing equation 2.5 ensures the system performs well on all such populations. In our toy example, minimizing equation 2.5 implies learning a classifier that is insensitive to perturbations along the horizontal (i.e. sensitive) direction. In Figure 1c this is achieved by the algorithm we describe next.

To keep things simple, we assume the hypothesis class is parametrized by $\theta \in \Theta \subset \mathbf{R}^d$ and replace the minimization with respect to $\mathcal{H}$ by minimization with respect to $\theta$. In light of the similarities between the DRO objective function and adversarial training, we borrow algorithms for adversarial training (Madry et al., 2017) to solve equation 2.5 (see Algorithm 2).

---

**Algorithm 2** Sensitive Subspace Robustness (SenSR)

---

**Require:** starting point $\hat{\theta}_1$, step sizes $\alpha_t, \beta_t > 0$
1: **repeat**
2: $\quad$ sample mini-batch $(x_1, y_1), \ldots, (x_B, y_B) \sim P_n$
3: $\quad x_{t_b}^* \leftarrow \arg\max_{x \in \mathcal{X}} \ell((x, y_{t_b}), \theta) - \hat{\lambda}_t d_x(x_{t_b}, x), b \in [B]$
4: $\quad \hat{\lambda}_{t+1} \leftarrow \max\{0, \hat{\lambda}_t - \alpha_t(\epsilon - \frac{1}{B}\sum_{b=1}^B d_x(x_{t_b}, x_{t_b}^*))\}$
5: $\quad \hat{\theta}_{t+1} \leftarrow \hat{\theta}_t - \frac{\beta_t}{B}\sum_{b=1}^B \partial_\theta \ell((x_{t_b}^*, y_{t_b}), \hat{\theta}_t)$
6: **until** converged

---

**Related work** Our approach to fair training is an instance of distributionally robust optimization (DRO). In DRO, the usual sample-average approximation of the expected cost function is replaced by $\widehat{L}_{\text{DRO}}(\theta) \triangleq \sup_{P \in \mathcal{U}} \mathbb{E}_P\big[\ell(Z, \theta)\big]$, where $\mathcal{U}$ is a (data dependent) uncertainty set of probability distributions. The uncertainty set may be defined by moment or support constraints (Chen et al., 2007; Delage & Ye, 2010; Goh & Sim, 2010), $f$-divergences (Ben-Tal et al., 2012; Lam & Zhou,

2015; Miyato et al., 2015; Namkoong & Duchi, 2016), and Wasserstein distances (Shafieezadeh-Abadeh et al., 2015; Blanchet et al., 2016; Esfahani & Kuhn, 2015; Lee & Raginsky, 2017; Sinha et al., 2017). Most similar to our work is Hashimoto et al. (2018): they show that DRO with a $\chi^2$-neighborhood of the training data prevents representation disparity, i.e. minority groups tend to suffer higher losses because the training algorithm ignores them. One advantage of picking a Wasserstein uncertainty set is the set depends on the geometry of the sample space. This allows us to encode the correct notion of individual fairness for the ML task at hand in the Wasserstein distance.

Our approach to fair training is also similar to adversarial training (Madry et al., 2017), which hardens ML models against adversarial attacks by minimizing adversarial losses of the form $\sup_{u \in \mathcal{U}} \ell(z + u, \theta)$, where $\mathcal{U}$ is a set of allowable perturbations (Szegedy et al., 2013; Goodfellow et al., 2014; Papernot et al., 2015; Carlini & Wagner, 2016; Kurakin et al., 2016). Typically, $\mathcal{U}$ is a scaled $\ell_p$-norm ball: $\mathcal{U} = \{u : \|u\|_p \leq \epsilon\}$. Most similar to our work is Sinha et al. (2017): they consider an uncertainty set that is a Wasserstein neighborhood of the training data.

There are a few papers that consider adversarial approaches to algorithmic fairness. Zhang et al. (2018) propose an adversarial learning method that enforces equalized odds in which the adversary learns to predict the protected attribute from the output of the classifier. Edwards & Storkey (2015) propose an adversarial method for learning classifiers that satisfy demographic parity. Madras et al. (2018) generalize their method to learn classifiers that satisfy other (group) notions of algorithmic fairness. Garg et al. (2019) propose to use adversarial logit pairing (Kannan et al., 2018) to achieve fairness in text classification using a pre-specified list of counterfactual tokens.

## 3 SENSR TRAINS INDIVIDUALLY FAIR ML MODELS

One of the main benefits of our approach is it provably trains individually fair ML models. Further, it is possible for the learner to certify that an ML model is individually fair *a posteriori*. As we shall see, both are consequences of uniform convergence results for the DR loss class. More concretely, we study how quickly the uniform convergence error

$$\delta_n \triangleq \sup_{\theta \in \Theta} \left\{ \left| \sup_{P : W_*(P, P_n) \leq \epsilon} \mathbb{E}_P\big[\ell(Z, \theta)\big] - \sup_{P : W(P, P_n) \leq \epsilon} \mathbb{E}_P\big[\ell(Z, \theta)\big] \right| \right\}, \qquad (3.1)$$

where $W_*$ is the Wasserstein distance on $\Delta(\mathcal{Z})$ with a transportation cost function $c_*$ that is possibly different from $c$, vanishes. We permit some discrepancy in the (transportation) cost function to study the effect of a data-driven choice of $c$. In the rest of this section, we regard $c_*$ as the exact cost function and $c$ as a cost function learned from human supervision. We start by stating our assumptions on the ML task:

(A1) the feature space $\mathcal{X}$ is bounded: $D \triangleq \max\{\text{diam}(\mathcal{X}), \text{diam}_*(\mathcal{X})\} < \infty$;

(A2) the functions in the loss class $\mathcal{L} = \{\ell(\cdot, \theta) : \theta \in \Theta\}$ are non-negative and bounded: $0 \leq \ell(z, \theta) \leq M$ for all $z \in \mathcal{Z}$ and $\theta \in \Theta$, and $L$-Lipschitz with respect to $d_x$:

$$\sup_{\theta \in \Theta}\{\sup_{(x_1, y), (x_2, y) \in \mathcal{Z}} |\ell((x_1, y), \theta) - \ell((x_2, y), \theta)|\} \leq L d_x(x_1, x_2);$$

(A3) the discrepancy in the (transportation) cost function is uniformly bounded:

$$\sup_{(x_1, y), (x_2, y) \in \mathcal{Z}} |c((x_1, y), (x_2, y)) - c_*((x_1, y), (x_2, y))| \leq \delta_c D^2.$$

Assumptions A1 and A2 are standard (see (Lee & Raginsky, 2017, Assumption 1, 2, 3)) in the DRO literature. We emphasize that the constant $L$ in Assumption A2 is **not** the constant $L$ in the definition of metric fairness; it may be much larger. Thus most models that satisfy the conditions of the loss class are not individually fair in a meaningful sense.

Assumption A3 deserves further comment. Under A1, A3 is mild. For example, if the exact fair metric is

$$d_x(x_1, x_2) = (x_1 - x_2)^T \Sigma_*(x_1 - x_2)^{\frac{1}{2}},$$

then the error in the transportation cost function is at most

$$|c((x_1, y), (x_2, y)) - c_*((x_1, y), (x_2, y))|$$
$$= |(x_1 - x_2)^T \Sigma(x_1 - x_2) - (x_1 - x_2)^T \Sigma_*(x_1 - x_2)|$$
$$\leq D^2 \frac{\|\Sigma - \Sigma_*\|_2}{\lambda_{\min}(\Sigma_*)},$$

We see that the error in the transportation cost function vanishes in the large-sample limit as long as $\Sigma$ is a consistent estimator of $\Sigma_*$.

We state the uniform convergence result in terms of the *entropy integral* of the loss class: $\mathfrak{C}(\mathcal{L}) = \int_0^\infty \sqrt{\log N_\infty(\mathcal{F}, r)} dr$, where $N_\infty(\mathcal{L}, r)$ as the $r$-covering number of the loss class in the uniform metric. The entropy integral is a measure of the complexity of the loss class.

**Proposition 3.1** (uniform convergence). *Under Assumptions A1–A3, equation 3.1 satisfies*

$$\delta_n \leq \frac{48\mathfrak{C}(\mathcal{L})}{\sqrt{n}} + \frac{48LD^2}{\sqrt{n\epsilon}} + \frac{L\delta_c D^2}{\sqrt{\epsilon}} + M(\frac{\log \frac{2}{t}}{2n})^{\frac{1}{2}} \tag{3.2}$$

*with probability at least $1 - t$.*

We note that Proposition 3.1 is similar to the generalization error bounds by Lee & Raginsky (2017). The main novelty in Proposition 3.1 is allowing error in the transportation cost function. We see that the discrepancy in the transportation cost function may affect the rate at which the uniform convergence error vanishes: it affects the rate if $\delta_c$ is $\omega_P(\frac{1}{\sqrt{n}})$.

A consequence of uniform convergence is SenSR trains individually fair classifiers (if there are such classifiers in the hypothesis class). By individually fair ML model, we mean an ML model that has a small gap

$$\sup_{P:W_*(P,P_*)\leq\epsilon} \mathbb{E}_P\big[\ell(Z, \theta)\big] - \mathbb{E}_{P_*}\big[\ell(Z, \theta)\big], \tag{3.3}$$

The gap is the difference between the optimal value of the auditor's optimization problem equation 2.1 and the (non-robust) risk. A small gap implies the auditor cannot significantly increase the loss by moving samples from $P_*$ to comparable samples.

**Proposition 3.2.** *Under the assumptions A1–A3, as long as there is $\bar{\theta} \in \Theta$ such that*

$$\sup_{P:W_*(P,P_*)\leq\epsilon} \mathbb{E}_P\big[\ell(Z, \bar{\theta})\big] \leq \delta^* \tag{3.4}$$

*for some $\delta^* > 0$, $\hat{\theta} \in \arg\min_{\theta\in\Theta} \sup_{P:W(P,P_n)\leq\epsilon} \mathbb{E}_P\big[\ell(Z, h)\big]$ satisfies*

$$\sup_{P:W_*(P,P_*)\leq\epsilon} \mathbb{E}_P\big[\ell(Z, \hat{\theta})\big] - \mathbb{E}_{P_*}\big[\ell(Z, \hat{\theta})\big] \leq \delta^* + 2\delta_n,$$

*where $\delta_n$ is the uniform convergence error equation 3.1.*

Proposition 3.2 guarantees Algorithm 2 trains an individually fair ML model. More precisely, if there are models in $\mathcal{H}$ that are (i) individually fair and (ii) achieve small test error, then Algorithm 2 trains such a model. It is possible to replace equation 3.4 with other conditions, but a condition to its effect cannot be dispensed with entirely. If there are no individually fair models in $\mathcal{H}$, then it is not possible for equation 2.5 to learn an individually fair model. If there are individually fair models in $\mathcal{H}$, but they all perform poorly, then the goal of learning an individually fair model is futile.

Another consequence of uniform convergence is equation 3.3 is close to its empirical counterpart

$$\sup_{P:W(P,P_n)\leq\epsilon} \mathbb{E}_P\big[\ell(Z, \theta)\big] - \mathbb{E}_{P_n}\big[\ell(Z, \theta)\big]. \tag{3.5}$$

In other words, the gap *generalizes*. This implies equation 3.5 is a *certificate of individual fairness*; *i.e.* it is possible for practitioners to check whether an ML model is individually fair by evaluating equation 3.5.

**Proposition 3.3.** *Under the assumptions A1–A3, for any $\epsilon > 0$,*

$$\sup_{\theta\in\Theta} \Big\{ \sup_{P:W(P,P_n)\leq\epsilon} \mathbb{E}_P\big[\ell(Z, \theta)\big] - \mathbb{E}_{P_n}\big[\ell(Z, \theta)\big] - (\sup_{P:W(P,P_*)\leq\epsilon} \mathbb{E}_P\big[\ell(Z, \theta)\big] - \mathbb{E}_{P_*}\big[\ell(Z, \theta)\big]) \Big\}$$
$$\leq 2\delta_n \text{ w.p. at least } 1 - t.$$

## 4 COMPUTATIONAL RESULTS

In this section, we present results from using SenSR to train individually fair ML models for two tasks: sentiment analysis and income prediction. We pick these two tasks to demonstrate the efficacy of SenSR on problems with structured (income prediction) and unstructured (sentiment analysis) inputs and in which the sensitive attribute (income prediction) is observed and unobserved (sentiment analysis). We refer to Appendix C and D for the implementation details.

Table 1: Sentiment prediction experiments over 10 restarts

|          | Acc.,%   | Race gap  | Gend. gap | Cuis. gap |
|----------|----------|-----------|-----------|-----------|
| SenSR    | 94±1     | 0.30±.05  | 0.19±.03  | **0.23±.05** |
| SenSR-E  | 93±1     | **0.11±.04** | **0.04±.03** | 1.11±.15  |
| Baseline | **95±1** | 7.01±.44  | 5.59±.37  | 4.10±.44  |
| Project  | 94±1     | 1.00±.56  | 1.99±.58  | 1.70±.41  |
| Sinha+   | 94±1     | 3.88±.26  | 1.42±.29  | 1.33±.18  |
| Bolukb.+ | 94±1     | 6.85±.53  | 4.33±.46  | 3.44±.29  |

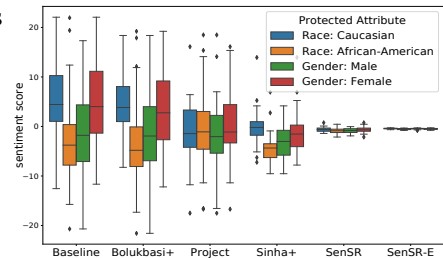

Figure 2: Box-plots of sentiment scores

### 4.1 FAIR SENTIMENT PREDICTION WITH WORD EMBEDDINGS

**Problem formulation** We study the problem of classifying the sentiment of words using positive (e.g. 'smart') and negative (e.g. 'anxiety') words compiled by Hu & Liu (2004). We embed words using 300-dimensional GloVe (Pennington et al., 2014) and train a one layer neural network with 1000 hidden units. Such classifier achieves 95% test accuracy, however it entails major individual fairness violation. Consider an application of this sentiment classifier to summarizing customer reviews, tweets or news articles. Human names are typical in such texts and should not affect the sentiment score, hence we consider fair metric between any pair of names to be 0. Then sentiment score for all names should be the same to satisfy the individual fairness. To make a connection to group fairness, following the study of Caliskan et al. (2017) that reveals the biases in word embeddings, we evaluate the fairness of our sentiment classifier using male and female names typical for Caucasian and African-American ethnic groups. We emphasize that to satisfy individual fairness, the sentiment of *any* name should be the same.

**Comparison metrics** To evaluate the gap between two groups of names, $\mathcal{N}_0$ for Caucasian (or female) and $\mathcal{N}_1$ for African-American (or male), we report $\frac{1}{|\mathcal{N}_0|}\sum_{n\in\mathcal{N}_0}(h(n)_1 - h(n)_0) - \frac{1}{|\mathcal{N}_1|}\sum_{n\in\mathcal{N}_1}(h(n)_1 - h(n)_0)$, where $h(n)_k$ is logits for class $k$ of name $n$ ($k = 1$ is the positive class). We use list of names provided in Caliskan et al. (2017), which consists of 49 Caucasian and 45 African-American names, among those 48 are female and 46 are male. The gap between African-American and Caucasian names is reported as Race gap, while the gap between male and female names is reported as Gend. gap in Table 1. As in Speer (2017), we also compare sentiment difference of two sentences: "Let's go get Italian food" and "Let's go get Mexican food", i.e. cuisine gap (abbreviated Cuis. gap in Table 1), as a test of generalization beyond names. To embed these sentences we average their word embeddings.

**Sensitive subspace** We consider embeddings of 94 names that we use for evaluation as sensitive directions, which may be regarded as utilizing the expert knowledge, i.e. these names form a list of words that an expert believes should be treated equally. Fair metric is then defined using an orthogonal complement projector of the span of sensitive directions as we discussed in Section 2.1. When expert knowledge is not available, or we wish to achieve general fairness for names, we utilize a side dataset of popular baby names in New York City.[1] The dataset has 11k names, however only 32 overlap with the list of names used for evaluation. Embeddings of these names define a group of comparable samples that we use to learn sensitive directions with SVD (see Appendix B.2 and Algorithm 3 for details). We take top 50 singular vectors to form the sensitive subspace. It is worth noting that, unlike many existing approaches in the fairness literature, we do not use any protected attribute information. Our algorithm only utilizes training words, their sentiments and a vanilla list of names.

**Results** From the box-plots in Figure 2, we see that both race and gender gaps are significant when using the baseline neural network classifier. It tends to predict Caucasian names as "positive", while the median for African-American names is negative; the median sentiment for female names is higher than that for male names. We considered three other approaches to this problem: the algorithm of Bolukbasi et al. (2016) for pre-processing word embeddings; pre-processing via projecting

---

[1] titled "Popular Baby Names" and available from `https://catalog.data.gov/dataset/`

Table 2: Summary of *Adult* classification experiments over 10 restarts

|  | B-Acc,% | S-Con. | GR-Con. | $\text{Gap}_G^{\text{RMS}}$ | $\text{Gap}_R^{\text{RMS}}$ | $\text{Gap}_G^{\text{max}}$ | $\text{Gap}_R^{\text{max}}$ |
|---|---|---|---|---|---|---|---|
| SenSR | 78.9 | **.934** | .984 | **.068** | **.055** | **.087** | **.067** |
| Baseline | **82.9** | .848 | .865 | .179 | .089 | .216 | .105 |
| Project | 82.7 | .868 | **1.00** | .145 | .064 | .192 | .086 |
| Adv. Debias. | 81.5 | .807 | .841 | .082 | .070 | .110 | .078 |
| CoCL | 79.0 | - | - | .163 | .080 | .201 | .109 |

out the sensitive subspace that we used for training SenSR (this is analogous to Prost et al. (2019)); training a distributionally robust classifier with Euclidean distance cost (Sinha et al., 2017). All approaches improved upon the baseline, however only SenSR can be considered individually fair. Our algorithm practically eliminates gender and racial gaps and achieves the notion of individual fairness as can be seen from almost equal predicted sentiment score for *all* names. We remark that using expert knowledge (i.e. evaluation names) allowed SenSR-E (E for expert) to further improve both group and individual fairness. However we warn practitioners that if the expert knowledge is too specific, generalization outside of the expert knowledge may not be very good. In Table 1 we report results averaged across 10 repetitions with 90%/10% train/test splits, where we also verify that accuracy trade-off with the baseline is minor. In the right column we present the generalization check, i.e. comparing a pair of sentences unrelated to names. Utilizing expert knowledge led to a fairness over-fitting effect, however we still see improvement over other methods. When utilizing SVD of a larger dataset of names we observe better generalization. Our generalization check suggests that fairness over-fitting is possible, therefore datasets and procedure for verifying fairness generalization are needed.

## 4.2 ADULT

**Problem formulation** Demonstrating the broad applicability of SenSR outside of natural language processing tasks, we apply SenSR to a classification task on the *Adult* (Dua & Graff, 2017) data set to predict whether an individual makes at least $50k based on features like gender and occupation for approximately 45,000 individuals. Models that predict income without fairness considerations can contribute to the problem of differences in pay between genders or races for the same work. Throughout this section, gender (male or female) and race (Caucasian or non-Caucasian) are binary.

**Comparison metrics** Arguably a classifier is individually unfair if the classifications for two data points that are the same on all features except demographic features are different. Therefore, to assess individual fairness, we report spouse consistency (S-Con.) and gender and race consistency (GR-Con.), which are measures of how often classifications change only because of differences in demographic features. For S-Con (resp. GR-con), we make 2 (resp. 4) copies of every data point where the only difference is that one is a husband and the other is a wife (resp. difference is in gender and race). S-Con (resp. GR-Con) is the fraction of corresponding pairs (resp. quadruples) that have the same classification. We also report various group fairness measures proposed by De-Arteaga et al. (2019) with respect to race or gender based on true positive rates, i.e. the ability of a classifier to correctly identify a given class. See Appendix D.5 for the definitions. We report $\text{Gap}_R^{\text{RMS}}$, $\text{Gap}_G^{\text{RMS}}$, $\text{Gap}_R^{\text{max}}$, and $\text{Gap}_G^{\text{max}}$ where $R$ refers to race, and $G$ refers to gender. We use balanced accuracy (B-acc) instead of accuracy[2] to measure predictive ability since only 25% of individuals make at least $50k.

**Sensitive subspace** Let $\{(x_i, x_{g_i})\}_{i=1}^m$ be the set of features $x_i \in \mathbb{R}^D$ of the data except the coordinate for gender is zeroed and where $x_{g_i}$ indicates the gender of individual $i$. For $\gamma > 0$, let $w_g = \arg\min_{w \in \mathbb{R}^D} \frac{1}{m} \sum_{i=1}^m -x_{g_i}(w^T x_i) + \log(1 + e^{w^T x_i}) + \gamma\|w\|_2$, i.e. $w_g$ is the learned hyperplane that classifies gender given by regularized logistic regression. Let $e_g \in \mathbb{R}^D$ (resp. $e_r$) be the vector that is 1 in the gender (resp. race) coordinate and 0 elsewhere. Then the sensitive subspace is the span of $[w_g, e_g, e_r]$. See Appendix B.1 for details.

---

[2]Accuracy is reported in Table 4 in Appendix D.

**Results**  See Table 2 for the average[3] of each metric on the test sets over ten 80%/20% train/test splits for Baseline, Project (projecting features onto the orthogonal complement of the sensitive subspace before training), CoCL (De-Arteaga et al., 2019), Adversarial Debiasing (Zhang et al., 2018), and SenSR. With the exception of CoCL (De-Arteaga et al., 2019), each classifier is a 100 unit single hidden layer neural network. The Baseline clearly exhibits individual and group fairness violations. While SenSR has the lowest B-acc, SenSR is the best by a large margin for S-Con. and has the best group fairness measures. We expect SenSR to do well on GR-consistency since the sensitive subspace includes the race and gender directions. However, SenSR's individually fair performance generalizes: the sensitive directions do not directly use the husband and wife directions, yet SenSR performs well on S-Con. Furthermore, SenSR outperforms Project on S-Con and group fairness measures illustrating that SenSR does much more than just ignoring the sensitive subspace. CoCL only barely improves group fairness compared to the baseline with a significant drop in B-acc and while Adversarial Debiasing also improves group fairness, it is worse than the baseline on individual fairness measures illustrating that group fairness does not imply individual fairness.

## 5   SUMMARY

We consider the task of training ML systems that are fair in the sense that their performance is invariant under certain perturbations in a sensitive subspace. This notion of fairness is a variant of individual fairness (Dwork et al., 2011). One of the main barriers to the adoption of individual fairness is the lack of consensus on a fair metric for many ML tasks. To circumvent this issue, we consider two approaches to learning a fair metric from data: one for problems in which the sensitive attribute is observed, and another for problems in which the sensitive attribute is unobserved. Given a data-driven choice of fair metric, we provide an algorithm that provably trains individually fair ML models.

ACKNOWLEDGMENTS

This work was supported by the National Science Foundation under grants DMS-1830247 and DMS-1916271.

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

# A  PROOFS

## A.1  PROOF OF PROPOSITION 3.1

By the duality result of Blanchet & Murthy (2016), for any $\epsilon > 0$,

$$
\sup_{P:W_*(P,P_*)\leq\epsilon} \mathbb{E}_P\big[\ell(Z,\theta)\big] - \sup_{P:W(P,P_n)\leq\epsilon} \mathbb{E}_P\big[\ell(Z,\theta)\big]
$$
$$
= \inf_{\lambda\geq 0}\big\{\lambda\epsilon + \mathbb{E}_{P_*}\big[\ell_\lambda^{c*}(Z,\theta)\big]\big\} - \lambda_n\epsilon + \mathbb{E}_{P_n}\big[\ell_{\lambda_n}^{c}(Z,\theta)\big]
$$
$$
\leq \mathbb{E}_{P_*}\big[\ell_{\lambda_n}^{c*}(Z,\theta)\big] - \mathbb{E}_{P_n}\big[\ell_{\lambda_n}^{c}(Z,\theta)\big],
$$

where $\lambda_n \in \arg\min_{\lambda\geq 0}\lambda\epsilon + \mathbb{E}_{P_n}\big[\ell_\lambda^{c}(Z,\theta)\big]$. By assumption A3,

$$
|\ell_{\lambda_n}^{c*}(z,\theta) - \ell_{\lambda_n}^{c}(z,\theta)|
$$
$$
= \left| \sup_{x_2\in\mathcal{X}} \ell((x_2,y),\theta) - \lambda_n c_*((x,y),(x_2,y)) - \sup_{x_2\in\mathcal{X}} \ell((x_2,y),\theta) - \lambda_n c((x,y),(x_2,y)) \right|
$$
$$
\leq \sup_{x_2\in\mathcal{X}} \lambda_n |c_*((x,y),(x_2,y)) - c((x,y),(x_2,y))|
$$
$$
\leq \lambda_n\delta_c \cdot D^2.
$$

This implies

$$
\sup_{P:W_*(P,P_*)\leq\epsilon} \mathbb{E}_P\big[\ell(Z,\theta)\big] - \sup_{P:W(P,P_n)\leq\epsilon} \mathbb{E}_P\big[\ell(Z,\theta)\big]
$$
$$
\leq \mathbb{E}_{P_*}\big[\ell_{\lambda_n}^{c*}(Z,\theta)\big] - \mathbb{E}_{P_n}\big[\ell_{\lambda_n}^{c*}(Z,\theta)\big] + \lambda_n\delta_c D^2.
$$

This bound is crude; it is possible to obtain sharper bounds under additional assumptions on the loss and transportation cost functions. We avoid this here to keep the result as general as possible.

Similarly,

$$
\sup_{P:W(P,P_n)\leq\epsilon} \mathbb{E}_P\big[\ell(Z,\theta)\big] - \sup_{P:W_*(P,P_*)\leq\epsilon} \mathbb{E}_P\big[\ell(Z,\theta)\big]
$$
$$
\leq \mathbb{E}_{P_n}\big[\ell_{\lambda_*}^{c}(Z,\theta)\big] - \mathbb{E}_{P_*}\big[\ell_{\lambda_*}^{c*}(Z,\theta)\big]
$$
$$
\leq \mathbb{E}_{P_n}\big[\ell_{\lambda_*}^{c*}(Z,\theta)\big] - \mathbb{E}_{P_*}\big[\ell_{\lambda_*}^{c*}(Z,\theta)\big] + \lambda_*\delta_c D^2,
$$

where $\lambda_* \in \arg\min_{\lambda\geq 0}\{\lambda\epsilon + \mathbb{E}_{P_*}\big[\ell_\lambda^{c*}(Z,\theta)\big]\}$.

**Lemma A.1** (Lee & Raginsky (2017)). *Let* $\tilde{\lambda} \in \arg\min_{\lambda\geq 0}\lambda\epsilon + \mathbb{E}_P\big[\ell_\lambda^{c}(Z,\theta)\big]$. *As long as the function in the loss class are $L$-Lipschitz with respect to $d_x$ (see Assumption A2), $\tilde{\lambda} \leq \frac{L}{\sqrt{\epsilon}}$.*

*Proof.* By the optimality of $\tilde{\lambda}$,

$$
\tilde{\lambda}\epsilon \leq \tilde{\lambda}\epsilon + \mathbb{E}_P\big[ \sup_{x_2\in\mathcal{X}} \ell((x_2,Y),\theta) - \tilde{\lambda}d_x(X,x_2)^2 - \ell((X,Y),\theta)\big]
$$
$$
= \tilde{\lambda}\epsilon + \mathbb{E}_P\big[\ell_{\tilde{\lambda}}^{c}(Z,\theta) - \ell(Z,\theta)\big]
$$
$$
\leq \lambda\epsilon + \mathbb{E}_P\big[\ell_\lambda^{c}(Z,\theta) - \ell(Z,\theta)\big]
$$
$$
= \lambda\epsilon + \mathbb{E}_P\big[ \sup_{x_2\in\mathcal{X}} \ell((x_2,Y),\theta) - \ell((X,Y),\theta) - \lambda d_x(X,x_2)^2\big]
$$

for any $\lambda \geq 0$. By Assumption A2, the right side is at most

$$
\tilde{\lambda}\epsilon \leq \lambda\epsilon + \mathbb{E}_P\big[ \sup_{x_2\in\mathcal{X}} Ld_x(X,x_2) - \lambda d_x(X,x_2)^2\big]
$$
$$
\leq \lambda\epsilon + \sup_{t\geq 0} Lt - \lambda t^2.
$$

We minimize the right side WRT $t$ (set $t = \frac{L}{2\lambda}$) and $\lambda$ (set $\lambda = \frac{L}{2\sqrt{\epsilon}}$) to obtain $\tilde{\lambda}\epsilon \leq L\sqrt{\epsilon}$. $\qquad\square$

By Lemma A.1, we have

$$\sup_{P:W_*(P,P_*)\leq\epsilon} \mathbb{E}_P\big[\ell(Z,\theta)\big] - \sup_{P:W(P,P_n)\leq\epsilon} \mathbb{E}_P\big[\ell(Z,\theta)\big] \leq \mathbb{E}_{P_*}\big[\ell_{\lambda_n}^{c*}(Z,\theta)\big] - \mathbb{E}_{P_n}\big[\ell_{\lambda_n}^{c*}(Z,\theta)\big] + \frac{L\delta_c D^2}{\sqrt{\epsilon}}$$

$$\sup_{P:W(P,P_n)\leq\epsilon} \mathbb{E}_P\big[\ell(Z,\theta)\big] - \sup_{P:W_*(P,P_*)\leq\epsilon} \mathbb{E}_P\big[\ell(Z,\theta)\big] \leq \mathbb{E}_{P_n}\big[\ell_{\lambda_*}^{c*}(Z,\theta)\big] - \mathbb{E}_{P_*}\big[\ell_{\lambda_*}^{c*}(Z,\theta)\big] + \frac{L\delta_c D^2}{\sqrt{\epsilon}}.$$

We combine the preceding bounds to obtain

$$\left| \sup_{P:W(P,P_n)\leq\epsilon} \mathbb{E}_P\big[\ell(Z,\theta)\big] - \sup_{P:W_*(P,P_*)\leq\epsilon} \mathbb{E}_P\big[\ell(Z,\theta)\big] \right|$$

$$\leq \sup_{f\in\mathcal{L}^{c*}} \big|\textstyle\int_{\mathcal{Z}} f(z)d(P_n - P_*)(z)\big| + \frac{L\delta_c D^2}{\sqrt{\epsilon}},$$

where $\mathcal{L}^{c*} = \{\ell_\lambda^{c*}(\cdot,\theta) : \lambda \in [0, \frac{L}{\sqrt{\epsilon}}], \theta \in \Theta\}$ is the DR loss class. In the rest of the proof, we bound $\sup_{f\in\mathcal{L}^{c*}} \big| \int_{\mathcal{Z}} f(z)d(P_* - P_n)(z)\big|$ with standard techniques from statistical learning theory. Assumption A2 implies the functions in $\mathcal{F}$ are bounded:

$$0 \leq \ell((x_1,y_1),\theta) - \underbrace{\lambda d_x(x_1,x_1)} \leq \ell_\lambda^c(z_1,\theta) \leq \sup_{x_2\in\mathcal{X}} \ell((x_2,y_1),\theta) \leq M.$$

This implies has bounded differences, so $\delta_n$ concentrates sharply around its expectation. By the bounded-differences inequality and a symmetrization argument,

$$\sup_{f\in\mathcal{L}^{c*}} \big|\textstyle\int_{\mathcal{Z}} f(z)d(P_n - P_*)(z)\big| \leq 2\mathfrak{R}_n(\mathcal{L}^{c*}) + M\big(\frac{\log\frac{2}{t}}{2n}\big)^{\frac{1}{2}}$$

WP at least $1 - t$, where $\mathfrak{R}_n(\mathcal{F})$ is the Rademacher complexity of $\mathcal{F}$:

$$\mathfrak{R}_n(\mathcal{F}) = \mathbb{E}\left[ \sup_{f\in\mathcal{F}} \frac{1}{n}\sum_{i=1}^n \sigma_i f(Z_i) \right].$$

**Lemma A.2.** *The Rademacher complexity of the DR loss class is at most*

$$\mathfrak{R}_n(\mathcal{L}^c) \leq \frac{24\mathfrak{C}(\mathcal{L})}{\sqrt{n}} + \frac{24LD^2}{\sqrt{n\epsilon}}.$$

*Proof.* To study the Rademacher complexity of $\mathcal{L}^c$, we first show that the $\mathcal{L}^c$-indexed Rademacher process $X_f \triangleq \frac{1}{n}\sum_{i=1}^n \sigma_i f(Z_i)$ is sub-Gaussian WRT to a pseudometric. Let $f_1 = \ell_{\lambda_1}^c(\cdot,\theta_1)$ and $f_2 = \ell_{\lambda_2}^c(\cdot,\theta_2)$. Define

$$d_{\mathcal{L}^c}(f_1, f_2) \triangleq \|\ell(\cdot,\theta_1) - \ell(\cdot,\theta_2)\|_\infty + D^2|\lambda_1 - \lambda_2|.$$

We check that $X_f$ is sub-Gaussian WRT $d_{\mathcal{L}^c}$:

$$\mathbb{E}\big[\exp(t(X_{f_1} - X_{f_2}))\big]$$

$$= \mathbb{E}\big[\exp\big(\frac{t}{n}\sum_{i=1}^n \sigma_i(\ell_{\lambda_1}^c(Z_i,\theta_1) - \ell_{\lambda_2}^c(Z_i,\theta_2))\big)\big]$$

$$= \mathbb{E}\big[\exp\big(\frac{t}{n}\sigma(\ell_{\lambda_1}^c(Z,\theta_1) - \ell_{\lambda_2}^c(Z,\theta_2))\big)\big]^n$$

$$= \mathbb{E}\big[\exp\big(\frac{t}{n}\sigma(\sup_{x_1\in\mathcal{X}}\inf_{x_2\in\mathcal{X}} \ell((x_1,Y),\theta_1) - \lambda_1 d_x(x_1,X)^2 - \ell((x_2,Y),\theta_2) + \lambda_2 d_x(X,x_2)^2)\big)\big]^n$$

$$= \mathbb{E}\big[\exp\big(\frac{t}{n}\sigma(\sup_{x_1\in\mathcal{X}} \ell((x_1,Y),\theta_1) - \ell((x_1,Y),\theta_2) + (\lambda_2 - \lambda_1)d_x(x_1,X)^2)\big)\big]^n$$

$$\leq \exp\big(\tfrac{1}{2}t^2 d_{\mathcal{L}^c}(f_1,f_2)\big).$$

Let $N(\mathcal{L}^c, d_{\mathcal{L}^c}, \epsilon)$ be the $\epsilon$-covering number of $(\mathcal{L}^c, d_{\mathcal{L}^c})$. We observe

$$N(\mathcal{L}^c, d_{\mathcal{L}^c}, \epsilon) \leq N(\mathcal{L}, \|\cdot\|_\infty, \tfrac{\epsilon}{2}) \cdot N([0, \tfrac{L}{\sqrt{\epsilon}}], |\cdot|, \tfrac{\epsilon}{2D^2}) \tag{A.1}$$

By Dudley's entropy integral,

$$
\begin{aligned}
\mathfrak{R}_n(\mathcal{L}^c) &\leq \frac{12}{\sqrt{n}} \int_0^\infty \log N(\mathcal{L}^c, d_{\mathcal{L}^c}, \epsilon)^{\frac{1}{2}} d\epsilon \\
&\leq \frac{12}{\sqrt{n}} \int_0^\infty \big( \log N(\mathcal{L}, \|\cdot\|_\infty, \tfrac{\epsilon}{2}) + N([0, \tfrac{L}{\sqrt{\epsilon}}], |\cdot|, \tfrac{\epsilon}{2D^2}) \big)^{\frac{1}{2}} d\epsilon \\
&\leq \frac{12}{\sqrt{n}} \bigg( \int_0^\infty \log N(\mathcal{L}, \|\cdot\|_\infty, \tfrac{\epsilon}{2})^{\frac{1}{2}} d\epsilon + \int_0^\infty N([0, \tfrac{L}{\sqrt{\epsilon}}], |\cdot|, \tfrac{\epsilon}{2D^2})^{\frac{1}{2}} d\epsilon \bigg) \\
&\leq \frac{24 \mathfrak{C}(\mathcal{L})}{\sqrt{n}} + \frac{24 L D^2}{\sqrt{n}\epsilon} \int_0^{\frac{1}{2}} \log(\tfrac{1}{\epsilon}) d\epsilon
\end{aligned}
$$

where we recalled equation A.1 in the second step. We evalaute the integral on the right side to arrive at the stated bound: $\int_0^{\frac{1}{2}} \log(\frac{1}{\epsilon}) d\epsilon < 1$. $\qquad\square$

By Lemma A.2,

$$
\sup_{f \in \mathcal{L}^{c*}} \big| \int_{\mathcal{Z}} f(z) d(P_n - P_*)(z) \big| \leq \frac{48 \mathfrak{C}(\mathcal{L})}{\sqrt{n}} + \frac{48 L D^2}{\sqrt{n}\epsilon} + M\big(\frac{\log \frac{2}{t}}{2n}\big)^{\frac{1}{2}},
$$

which implies

$$
\begin{aligned}
&\Big| \sup_{P:W(P,P_n) \leq \epsilon} \mathbb{E}_P\big[\ell(Z, \theta)\big] - \sup_{P:W_*(P,P_*) \leq \epsilon} \mathbb{E}_P\big[\ell(Z, \theta)\big] \Big| \\
&\qquad \leq \frac{48 \mathfrak{C}(\mathcal{L})}{\sqrt{n}} + \frac{48 L D^2}{\sqrt{n}\epsilon} + \frac{L \delta_c D^2}{\sqrt{\epsilon}} + M\big(\frac{\log \frac{2}{t}}{2n}\big)^{\frac{1}{2}}.
\end{aligned}
$$

WP at least $1 - t$.

## A.2 PROOFS OF PROPOSITIONS 3.2 AND 3.3

*Proof of Proposition 3.2.* It is enough to show

$$
\sup_{P:W_*(P,P_*) \leq \epsilon} \mathbb{E}_P\big[\ell(Z, \hat{\theta})\big] \leq \delta^* + 2\delta_n
$$

because the loss function is non-negative. We have

$$
\begin{aligned}
\sup_{P:W_*(P,P_*) \leq \epsilon} \mathbb{E}_P\big[\ell(Z, \hat{\theta})\big] &\leq \sup_{P:W(P,P_n) \leq \epsilon} \mathbb{E}_P\big[\ell(Z, \hat{\theta})\big] + \delta_n \\
&\leq \sup_{P:W(P,P_n) \leq \epsilon} \mathbb{E}_P\big[\ell(Z, \bar{\theta})\big] + \delta_n \\
&\leq \sup_{P:W_*(P,P_*) \leq \epsilon} \mathbb{E}_P\big[\ell(Z, \bar{\theta})\big] + 2\delta_n \\
&\leq \delta^* + 2\delta_n.
\end{aligned}
$$

$\qquad\square$

*Proof of Proposition 3.3.*

$$
\begin{aligned}
&\sup_{P:W_*(P,P_n) \leq \epsilon} \big( \mathbb{E}_P\big[\ell(Z, \theta)\big] - \mathbb{E}_{P_n}\big[\ell(Z, \theta)\big] \big) - \sup_{P:W(P,P_*) \leq \epsilon} \big( \mathbb{E}_P\big[\ell(Z, \theta)\big] - \mathbb{E}_{P_*}\big[\ell(Z, \theta)\big] \big) \\
&= \sup_{P:W_*(P,P_*) \leq \epsilon} \mathbb{E}_P\big[\ell(Z, \theta)\big] - \sup_{P:W(P,P_n) \leq \epsilon} \mathbb{E}_P\big[\ell(Z, \theta)\big] + \mathbb{E}_{P_*}\big[\ell(Z, \theta)\big] - \mathbb{E}_{P_n}\big[\ell(Z, \theta)\big] \\
&\leq \delta_n + \mathbb{E}_{P_*}\big[\ell(Z, \theta)\big] - \mathbb{E}_{P_n}\big[\ell(Z, \theta)\big]
\end{aligned}
$$

The loss function is bounded, so it is possible to bound $\mathbb{E}_{P_*}\big[\ell(Z, \theta)\big] - \mathbb{E}_{P_n}\big[\ell(Z, \theta)\big]$ by standard uniform convergence results on bounded loss classes. $\qquad\square$

## B    DATA-DRIVEN FAIR METRICS

### B.1    LEARNING THE FAIR METRIC FROM OBSERVATIONS OF THE SENSITIVE ATTRIBUTE

Here we assume the sensitive attribute is discrete and is observed for a small subset of the training data. Formally, we assume this subset of the training data has the form $\{(X_i, K_i, Y_i)\}$, where $K_i$ is the sensitive attribute of the $i$-th subject. To learn the sensitive subspace, we fit a softmax regression model to the data

$$\mathbb{P}(K_i = l \mid X_i) = \frac{\exp(a_l^T X_i + b_l)}{\sum_{l=1}^k \exp(a_l^T X_i + b_l)}, \; l = 1, \dots, k,$$

and take the span of $A = [a_1 \dots a_k]$ as the sensitive subspace to define the fair metric as

$$d_x(x_1, x_2)^2 = (x_1 - x_2)^T (I - P_{\mathsf{ran}(A)})(x_1 - x_2). \tag{B.1}$$

This approach readily generalizes to sensitive attributes that are not discrete-valued: replace the softmax model by an appropriate generalized linear model.

In many applications, the sensitive attribute is part of a user's demographic information, so it may not be available due to privacy restrictions. This does not preclude the proposed approach because the sensitive attribute is only needed to learn the fair metric and is neither needed to train the classifier nor at test time.

### B.2    LEARNING THE FAIR METRIC FROM COMPARABLE SAMPLES

In this section, we consider the task of learning a fair metric from supervision in a form of comparable samples. This type of supervision has been considered in the literature on debiasing learned representations. For example, method of Bolukbasi et al. (2016) for removing gender bias in word embeddings relies on sets of words whose embeddings mainly vary in a gender subspace (*e.g.* (king, queen)).

To keep things simple, we focus on learning a generalized Mahalanobis distance

$$d_x(x_1, x_2) = (\varphi(x_1) - \varphi(x_2))^T \widehat{\Sigma}(\varphi(x_1) - \varphi(x_2))^{\frac{1}{2}}, \tag{B.2}$$

where $\varphi(x) : \mathcal{X} \to \mathbf{R}^d$ is a *known* feature map and $\widehat{\Sigma} \in \mathbf{S}_+^{d \times d}$ is a covariance matrix. Our approach is based on a factor model

$$\varphi_i = A_* u_i + B_* v_i + \epsilon_i,$$

where $\varphi_i \in \mathbf{R}^d$ is the learned representation of $x_i$, $u_i \in \mathbf{R}^K$ (resp. $v_i \in \mathbf{R}^L$) is the sensitive/irrelevant (resp. relevant) attributes of $x_i$ to the task at hand, and $\epsilon_i$ is an error term. For example, in Bolukbasi et al. (2016), the learned representations are the embeddings of words in the vocabulary, and the sensitive attribute is the gender bias of the words. The sensitive and relevant attributes are generally unobserved.

Recall our goal is to obtain $\widehat{\Sigma}$ so that equation B.2 is small whenever $v_1 \approx v_2$. One possible choice of $\widehat{\Sigma}$ is the projection matrix onto the orthogonal complement of $\mathsf{ran}(A)$, which we denote by $P_{\mathsf{ran}(A)}$. Indeed,

$$d_x(x_1, x_2)^2 = (\varphi_1 - \varphi_2)^T (I - P_{\mathsf{ran}(A)})(\varphi_1 - \varphi_2) \tag{B.3}$$

$$\approx (v_1 - v_2)^T B_*^T (I - P_{\mathsf{ran}(A)}) B_* (v_1 - v_2), \tag{B.4}$$

which is small whenever $v_1 \approx v_2$. Although $\mathsf{ran}(A)$ is unknown, it is possible to estimate it from the learned representations and groups of comparable samples by factor analysis.

The factor model attributes variation in the learned representations to variation in the sensitive and relevant attributes. We consider two samples comparable if their relevant attributes are similar. In other words, if $\mathcal{I} \subset [n]$ is (the indices of) a group of comparable samples, then

$$H\Phi_{\mathcal{I}} = HU_{\mathcal{I}}A_*^T + \overset{\approx \, 0}{\underset{}{\cancel{HV_{\mathcal{I}}B_*^T}}} + HE_{\mathcal{I}} \approx HU_{\mathcal{I}}A_*^T + HE_{\mathcal{I}}, \tag{B.5}$$

where $H = I_{|\mathcal{I}|} - \frac{1}{|\mathcal{I}|} 1_{|\mathcal{I}|} 1_{|\mathcal{I}|}^T$ is the centering or de-meaning matrix and the rows of $\Phi_{\mathcal{I}}$ (resp. $U_{\mathcal{I}}$, $V_{\mathcal{I}}$) are $\varphi_i$ (resp. $u_i, v_i$). If this group of samples have identical relevant attributes, *i.e.* $V_{\mathcal{I}} = 1_{|\mathcal{I}|} v^T$ for some $v$, then $HV_{\mathcal{I}}$ vanishes exactly. As long as $u_i$ and $\epsilon_i$ are uncorrelated (*e.g.* $\mathbb{E}\big[u_i \epsilon_i^T\big] = 0$), equation B.5 implies

$$\mathbb{E}\big[\Phi_{\mathcal{I}}^T H \Phi_{\mathcal{I}}\big] \approx A\mathbb{E}\big[U_{\mathcal{I}}^T H U_{\mathcal{I}}\big] A^T + \mathbb{E}\big[E_{\mathcal{I}}^T H E_{\mathcal{I}}\big],$$

This suggests estimating $\mathrm{ran}(A)$ from the learned representations and groups of comparable samples by factor analysis. We summarize our approach in Algorithm 3.

---

**Algorithm 3** estimating $\widehat{\Sigma}$ for the fair metric

---

1: **Input:** $\{\varphi_i\}_{i=1}^n$, comparable groups $\mathcal{I}_1, \ldots, \mathcal{I}_G$
2: $\widehat{A}^T \in \arg\min_{W_g, A}\{\frac{1}{2}\sum_{g=1}^G \|H_g \Phi_{\mathcal{I}_g} - W_g A^T\|_F^2\}$        $\triangleright$ factor analysis
3: $Q \leftarrow \mathrm{qr}(\widehat{A})$        $\triangleright$ get orthonormal basis of $\mathrm{ran}(\widehat{A})$
4: $\widehat{\Sigma} \leftarrow I_d - QQ^T$

---

## C    SenSR implementation details

This section is to accompany the implementation of the SenSR algorithm and is best understood by reading it along with the code implemented using TensorFlow.[4] We discuss choices of learning rates and few specifics of the code. Words in *italics* correspond to variables in the code and following notation in parentheses defines corresponding name in Table 3, where we summarize all hyperparameter choices.

**Handling class imbalance**    Datasets we study have imbalanced classes. To handle it, on every *epoch*($E$) (i.e. number of epochs) we subsample a *batch_size*($B$) training samples enforcing equal number of observations per class. This procedure can be understood as data augmentation.

**Perturbations specifics**    Our implementation of SenSR algorithm has two inner optimization problems — subspace perturbation and full perturbation (when $\epsilon > 0$). Subspace perturbation can be viewed as an initialization procedure for the attack. We implement both using Adam optimizer (Kingma & Ba, 2014) inside the computation graph for better efficiency, i.e. defining corresponding perturbation parameters as Variables and re-setting them to zeros after every epoch. This is in contrast with a more common strategy in the adversarial robustness implementations, where perturbations (i.e. attacks) are implemented using tf.gradients with respect to the input data defined as a Placeholder.

**Learning rates**    As mentioned above, in addition to regular Adam optimizer for learning the parameters we invoke two more for the inner optimization problems of SenSR. We use same learning rate of 0.001 for the parameters optimizer, however different learning rates across datasets for *subspace_step*($s$) and *full_step*($f$). Two other related parameters are number of steps of the inner optimizations: *subspace_epoch*($se$) and *full_epoch*($fe$). We observed that setting subspace perturbation learning rate too small may prevent our algorithm from reducing unfairness, however setting it big does not seem to hurt. On the other hand, learning rate for full perturbation should not be set too big as it may prevent algorithm from solving the original task. Note that full perturbation learning rate should be smaller than perturbation budget *eps*($\epsilon$) — we always use $\epsilon/10$. In general, malfunctioning behaviors are immediately noticeable during training and can be easily corrected, therefore we did not need to use any hyperparameter optimization tools.

---

[4]https://github.com/IBM/sensitive-subspace-robustness

Table 3: SenSR hyperparameter choices in the experiments

|           | $E$  | $B$ | $s$  | $se$ | $\epsilon$ | $f$       | $fe$ |
| --------- | ---- | --- | ---- | ---- | ---------- | --------- | ---- |
| Sentiment | 4K   | 1K  | 0.1  | 10   | 0.1        | 0.01      | 10   |
| Adult     | 12K  | 1K  | 10   | 50   | $10^{-3}$  | $10^{-4}$ | 40   |

Table 4: Summary of *Adult* classification experiments over 10 restarts

|              | Accuracy      | B-TPR         | $\mathrm{Gap}_G^{\mathrm{RMS}}$ | $\mathrm{Gap}_R^{\mathrm{RMS}}$ | $\mathrm{Gap}_G^{\max}$ | $\mathrm{Gap}_R^{\max}$ |
| ------------ | ------------- | ------------- | ------------------------------- | ------------------------------- | ----------------------- | ----------------------- |
| SenSR        | .787±.003     | .789±.003     | **.068**±.004                   | **.055**±.003                   | **.087**±.005           | **.067**±.004           |
| Baseline     | **.813**±.001 | **.829**±.001 | .179±.004                       | .089±.003                       | .216±.003               | .105±.003               |
| Project      | **.813**±.001 | .827±.001     | .145±.004                       | .064±.003                       | .192±.004               | .086±.004               |
| Adv. Debias. | .812±.001     | .815±.002     | .082±.005                       | .070±.006                       | .110±.006               | .078±.005               |
| CoCL         | -             | .790          | .163                            | .080                            | .201                    | .109                    |

# D   ADDITIONAL ADULT EXPERIMENT DETAILS

## D.1   PREPROCESSING

The continuous features in *Adult* are the following: `age`, `fnlwgt`, `capital-gain`, `capital-loss`, `hours-per-week`, and `education-num`. The categorical features are the following: `workclass`, `education`, `marital-stataus`, `occupation`, `relationship`, `race`, `sex`, `native-country`. See Dua & Graff (2017) for a description of each feature. We remove `fnlwgt` and `education` but keep `education-num`, which is a integer representation of education. We do not use `native-country`, but use `race` and `sex` as predictive features. We treat `race` as binary: individuals are either White or non-White. For every categorical feature, we use one hot encoding. For every continuous feature, we standardize, i.e., subtract the mean and divide by the standard deviation. We remove anyone with missing data leaving 45,222 individuals.

This data is imbalanced: 25% make at least $50k per year. Furthermore, there is demographic imbalance with respect to race and gender as well as class imbalance on the outcome when conditioning on race or gender: 86% of individuals are white of which 26% make at least $50k a year; 67% of individuals are male of which 31% make at least $50k a year; 11% of females make at least $50k a year; and 15% of non-whites make at least $50k a year.

## D.2   FULL EXPERIMENTAL RESULTS

See Tables 4 and 5 for the full experiment results. The tables report the average and the standard error for each metric on the test set for 10 train and test splits.

## D.3   SENSITIVE SUBSPACE

To learn the hyperplane that classifies females and males, we use our implementation of regularized logistic regression with a batch size of 5k, 5k epochs, and .1 $\ell_2$ regularization.

Table 5: Summary of individual fairness metrics in *Adult* classification experiments over 10 restarts

|              | Spouse Consistency | Gender and Race Consistency |
| ------------ | ------------------ | --------------------------- |
| SenSR        | **.934**±.012      | .984±.000                   |
| Baseline     | .848±.008          | .865±.004                   |
| Project      | .868±.005          | **1**±0                     |
| Adv. Debias. | .807±.002          | .841±.012                   |

### D.4 HYPERPARAMETERS AND TRAINING

For each model, we use the same 10 train/test splits where use 80% of the data for training. Because of the class imbalance, each minibatch is sampled so that there are an equal number of training points from both the "income at least $50k class" and the "income below $50k class."

#### D.4.1 BASELINE, PROJECT, AND SENSR

See Table 3 for the hyperparameters we used when training Baseline, Project, and SenSR (Baseline and Project use a subset). Hyperparameters are defined in Appendix C.

#### D.4.2 ADVESARIAL DEBIASING

We used Zhang et al. (2018)'s adversarial debiasing implementation in IBM's AIF360 package (Bellamy et al., 2018) where the source code was modified so that each mini-batch is balanced with respect to the binary labels just as we did with our experiments and dropout was not used. Hyperparameters are the following: adversary loss weight $= .001$, num epochs $= 500$, batch size $= 1000$, and privileged groups are defined by binary gender and binary race.

### D.5 GROUP FAIR METRICS

Let $\mathcal{C}$ be a set of classes, $A$ be a binary protected attribute and $Y, \hat{Y} \in \mathcal{C}$ be the true class label and the predicted class label. Then for $a \in \{0, 1\}$ and $c \in \mathcal{C}$ define $\text{TPR}_{a,c} = \mathbb{P}(\hat{Y} = c | A = a, Y = c)$; $\text{Gap}_{A,c} = \text{TPR}_{0,c} - \text{TPR}_{1,c}$; $\text{Gap}_A^{\text{RMS}} = \sqrt{\frac{1}{|\mathcal{C}|} \sum_{c \in \mathcal{C}} \text{Gap}_{A,c}^2}$; $\text{Gap}_A^{\max} = \arg\max_{c \in \mathcal{C}} |\text{Gap}_{A,c}|$; $\text{Balanced Acc} = \frac{1}{|\mathcal{C}|} \sum_{c \in \mathcal{C}} \mathbb{P}(\hat{Y} = c | Y = c)$.

For Adult, we report $\text{Gap}_R^{\text{RMS}}$, $\text{Gap}_G^{\text{RMS}}$, $\text{Gap}_R^{\max}$, and $\text{Gap}_G^{\max}$ where $\mathcal{C}$ is composed of the two classes that correspond to whether someone made at least $50k, $R$ refers to race, and $G$ refers to gender.

