# OpenReview forum: "Training individually fair ML models with sensitive subspace robustness"
_ICLR.cc/2020/Conference — Accept (Spotlight)_

### Official Review · AnonReviewer3 · 2019-10-22
**Official Blind Review #3**

**Rating:** 6

**Review:**

This paper proposes a new definition of algorithmic fairness that is based on the idea of individual fairness. They then present an algorithm that will provably find an ML model that satisfies the fairness constraint (if such a model exists in the search space). One needed ingredient for the fairness constraint is a distance function (or "metric") in the input space that captures the fact that some features should be irrelevant to the classification task. That is, under this distance function, input that differ only in sensitive attributes like race or gender should be close-by. The idea of the fairness constraint is that by perturbing the inputs (while keeping them close with respect to the distance function), the loss of the model cannot be significantly increased. Thus, this fairness constraint is very much related to robustness.

---

Overall, I like the basic idea of the paper but I found the presentation lacking.

I do think their idea for a fairness constraint is very interesting, but it gets too bogged down in the details of the mathematical theory. They mention Dwork et al. at the beginning but don't really compare it to their idea in detail, even though I think there would be a lot of interesting things to say about this. For example, the definition by Dwork et al. seems to imply that some labels in the training set might be incorrect, whereas the definition in this paper does not seem to imply that (which I think is a good thing).

The main problem in section 2 is that the choice of distance function is barely discussed although that's what's most important to make the result fair. For all the mathematical rigor in section 2, the paragraph that is arguing that the defined constraint encourages fairness is somewhat weak. Here a comparison to other fairness definitions and an in-depth discussion of the distance function would help.

(In general I felt that this part was more trying to impress the reader than trying to explain, but I will try to not hold it against this paper.)

As it is, I feel the paper cannot be completely understood without reading the appendix.

There is also this sentence at the bottom of page 5: "A small gap implies the investigator cannot significantly increase the loss by moving samples from $P_*$ to comparable samples." This should have been at the beginning of section 2 in order to motivate the derivation.

In the experiments, I'm not sure how useful the result of the word embedding experiment really is. Either someone is interested in the sentiment associated with names, in which case your method renders the predicted sentiments useless or someone is not interested in the sentiment associated with names and your method doesn't even have any effect.

Final point: while I like the idea of the balanced TPR, I think the name is a bit misleading because, for example, in the binary case it is the average of the TPR and the TNR. Did you invent this terminology? If so, might I suggest another name like balanced accuracy?

I would change the score (upwards) if the following things are addressed:

- make it easier to understand the main point of the paper
- make more of a comparison to Dwork et al. or other fairness definitions
- fix the following minor mistakes

Minor comments:

- page 2, beginning of section 2: you use the word "regulator" here once but everywhere else you use "investigator"
- equation 2.1: as far as I can tell $M$ is not defined anywhere; you might mean $\Delta (\mathcal{Z})$
- page 3, sentence before Eq 2.3: what does the $\#$ symbol mean?
- page 3, sentence before Eq 2.3: what is $T$? is it $T_\lambda$?
- Algorithm 2: what is the difference between $\lambda^*_t$ and $\hat{\lambda}_t$?
- page 7: you used a backslash between "90%" and "10%" and "train" and "test". That would traditionally be a normal slash.
- in appendix B: the explanation for what $P_{ran(A)}$ means should be closer to the first usage
- in the references, you list one paper twice (the one by Zhang et al.)

EDIT: changed the score after looking at the revised version

**Experience Assessment:**

I have published one or two papers in this area.

**Review Assessment: Checking Correctness Of Derivations And Theory:**

I assessed the sensibility of the derivations and theory.

**Review Assessment: Checking Correctness Of Experiments:**

I carefully checked the experiments.

**Review Assessment: Thoroughness In Paper Reading:**

I read the paper at least twice and used my best judgement in assessing the paper.

---

> ### Author Response · Authors · 2019-11-08
> **Response to Reviewer 3**
>
> Thank you for the feedback. We address the key issues you mentioned below and we have updated the draft accordingly.
>
> You are correct that the fairness constraint is not exactly Dwork et al's notion of individual fairness, but it is very similar. We added an explicit statement of our definition in section 2 (see (2.2)). We also added a passage to section 2 comparing the two notions. In summary, we modify Dwork et al's definition in two ways: (i) instead of requiring the output of the ML model to be similar on all inputs comparable to a training example, we require the output to be similar to the training label; (ii) we use the increase in loss value to measure the difference between the outputs of a predictor on the different training sets instead of a metric on the output space of the predictor. The main benefits of these modifications are (i) this modified notion of individual fairness encodes not only (individual) fairness but also accuracy (as you noted in your comments), (ii) it is possible to optimize the fairness constraint efficiently, (iii) we can show this modified notion of individual fairness generalizes (see section 3 for formal statements). The unfortunate side effect is the additional mathematical details.
>
> The detailed description of the metric is in the Appendix B. To help readers find the description, we added references to it where necessary. We also added a summary of how we learn the metric near the beginning of section 2.
>
> The resume screening example at the beginning of section 2 is our motivation for the subsequent derivations, we added a bit to the first paragraph of section 2 to make the connection between the example and the derivations clear.
>
> In the word embedding experiment the application we have in mind is when someone needs to evaluate sentiment of sentences that can contain negative/positive sentiment words and names at the same time. Sentiment of a sentence can be evaluated by averaging sentiments of the corresponding words. This application is motivated by the paper "Mining and summarizing customer reviews" of Hu, M. and Liu, B. (2004). Training and testing dataset of positive and negative words also originates in their paper. From the perspective of individual fairness, when summarizing customer reviews, our sentiment prediction for two hypothetical restaurant reviews "My friend Adam liked their pizza" and "My friend Tashika liked their pizza" should be the same. As our experiment shows, this is achieved with SenSR. Resulting classifier is good at identifying sentiment of words and does not discriminate against names at the same time. It also reduces discrimination beyond names, e.g. "Let’s go get Italian food" and "Let’s go get Mexican food" have almost identical sentiment prediction with SenSR and severely biased in favor of the Italian food when using baseline classifier.
>
> We borrowed the term balanced TPR from Romanov et al (2019), but we are not particular tied to the term. We changed all instances of balanced TPR to balanced accuracy.
>
> We corrected the minor mistakes you mentioned.
>
> Refs: Romanov et al, What's in a Name? Reducing Bias in Bios without Access to Protected Attributes, NAACL 2019.

---

### Official Review · AnonReviewer2 · 2019-10-24
**Official Blind Review #2**

**Rating:** 6

**Review:**


Summary
The authors propose training to optimize individual fairness using sensitive subspace robustness (SenSR) algorithm.

Decision
Overall, I recommend borderline as the paper seems legit in formulating the individual fairness problem into a minmax robust optimization problem. The authors show improvement in gender and racial biases compared to non-individual fair approaches. However, I think some sections are hard to follow for people not in the field.

Supporting argument:
1. End of P3, it is not clear to me why solving the worst case is better.
2. Though this paper studied individual fairness, can it also work for group fairness? I am not sure whether this is the only work in this direction (baselines are not for individual fairness).
3. Some of the metrics in the experiments are not precisely defined such as Race gap, Cuis. gap, S-Con, GR-Con. It is hard to follow from the text description.
4. Some baseline models are not clearly defined such as “Project” in Table 1.
5. Not sure how Section 3 connects with the rest of the paper.


Additional feedback:
1. Missing reference: https://arxiv.org/abs/1907.12059
2. What’s TV distance in introduction?


**Experience Assessment:**

I do not know much about this area.

**Review Assessment: Checking Correctness Of Derivations And Theory:**

I did not assess the derivations or theory.

**Review Assessment: Checking Correctness Of Experiments:**

I did not assess the experiments.

**Review Assessment: Thoroughness In Paper Reading:**

I made a quick assessment of this paper.

---

> ### Author Response · Authors · 2019-11-08
> **Response to Reviewer 2**
>
> Thank you for the feedback. We address your concerns below.
>
> 1. The objective that we minimize is the worst-case performance of a predictor on hypothetical training sets that are similar (only differ in irrelevant features) to the observed training set. This leads to fairness because it penalizes predictors that perform well on the observed training set but poorly on similar hypothetical training sets. For example, an unfair resume screening model may perform very well on a set of training resumes from mostly white men, but poorly on resumes from women or minorities. By considering hypothetical sets of resumes from women or minorities during training, the objective we minimize penalizes models that only perform well on white men.
>
> 2. You can certainly encode group fairness by picking a metric that declares a pair of inputs similar whenever they are from the same group, but this is tangential to our goal of operationalizing individual fairness. We have baselines and metrics for group fairness because group fairness is the prevalent notion in the literature.
>
> 3. Each of the experiments has a dedicated "Comparison metrics" paragraph. We clarified the definitions of race and gender gaps in the corresponding paragraph. They are the differences between average logits output by the classifier evaluated at Caucasian vs African-American names for the Race gap and Male vs Female names for the Gender gap. Cuisine gap is the difference between logits of the embedded sentences: "Let’s go get Italian food" and "Let’s go get Mexican food". Spouse Consistency (S-Con.) and Gender and Race Consistency (GR-Con.) quantify the individual fairness intuition, i.e. how often classifier prediction remains unchanged when we evaluate it on a hypothetical "counterfactual" example created by changing features such as gender and keeping all other features unchanged. For these individual fairness metrics we did not write mathematical definition, but are happy to add one if the reviewer believes it would improve clarity.
>
> 4. In our experiments we discuss all baselines in the corresponding "Results" paragraphs. Project is the pre-processing baseline where we project data onto the orthogonal complement of the sensitive subspace and then train regular classifier with the projected data. SenSR outperforms this baseline suggesting that simply projecting out sensitive subspace is not sufficient and that robustness to unfair perturbations through SenSR gives better results in terms of fairness. This is analogous to the observation made in the group fairness literature that simply excluding protected attribute is not sufficient to achieve fairness.
>
> 5. The main point of section 3 is to show that the fairness constraint generalizes; i.e. if you  train a model with SenSR, and it performs well on all hypothetical training sets that are similar to the observed training set (i.e. it seems fair on the training data), then it also performs well with high probability (WHP) on all hypothetical test sets that are similar to a test set (i.e. it is fair WHP at test time).
>
> We added the missing reference and clarified what is TV distance in the introduction.

---

### Official Review · AnonReviewer1 · 2019-10-28
**Official Blind Review #1**

**Rating:** 8

**Review:**

General:
The authors propose a method to train individually fair ML models by pursuing robustness of the similarity loss function among the comparable data points. The main algorithmic tool of training is borrowed from the recent adversarial training, and the paper also gives the theoretical analyses on the convergence property of their method.

Pros:
1. They make the point that the individual fairness is important.
2. The paper proposes a practical algorithm for achieving the robustness and the indivdual fairness. Formulating that the main criterion for checking the fainess is Eq.(2.1), the paper takes a sensible route of using dual and minimax optimization problem (2.4).
3. The experimental results are compelling – while the proposed method loses the accuracy a bit, but shows very good individual fairness under their used metric.

Cons & Questions:
1. What is the empirical convergence property of the algorithm? How long does it take to train for the experiments given?
2. It seems like the main tools for algorithm and theory are borrowed from other papers in adversarial training e.g., (Madry 2017). Are their any algorithmic alternatives for solving (2.4)?
3. Why do you use d_z^2 instead of d_z for defining c(z_1,z_2)?
4. What happens when you use more complex models than 1 layer neural net?

**Experience Assessment:**

I have read many papers in this area.

**Review Assessment: Checking Correctness Of Derivations And Theory:**

I assessed the sensibility of the derivations and theory.

**Review Assessment: Checking Correctness Of Experiments:**

I assessed the sensibility of the experiments.

**Review Assessment: Thoroughness In Paper Reading:**

I read the paper at least twice and used my best judgement in assessing the paper.

---

> ### Author Response · Authors · 2019-11-08
> **Response to Reviewer 1**
>
> Thank you for the feedback. We address the Cons & Questions in what follows.
>
> 1. On a laptop without GPU, training SenSR on the sentiment data (experiment in Section 4.1) takes about 6 minutes.
>
> 2. You are correct that the proposed algorithm is similar to adversarial training. We consider this a benefit of our approach because it allows practitioners to borrow algorithms for adversarial training to train fair ML models. Theoretically speaking, the main distinction of our approach is a generalization error bound for data-driven Wasserstein distributed robust optimization (DRO). In most prior work on Wasserstein DRO, the metric is known, so there is no need to study the effect of error in the metric on generalization. In our application, the metric is learned from data, and we show that generalization degrades gracefully with error in the metric (see the third term on the right side of (3.2)).
>
> 3. We use d_z^2 instead of d_z because it is a common choice in Wasserstein DRO. For example, Sinha et al also use the squared Euclidean distance.
>
> 4. To answer the question about more complex models, we trained a deep neural network with 10 hidden layers (100 neurons each) on the sentiment prediction task (using exactly same hyperparameters as in the paper). SenSR continues to be effective: test accuracy is 94.3% and race gap is 0.2.
>
> Refs: Sinha et al, Certifying Some Distributional Robustness with Principled Adversarial Training, ICLR 2018.

---

### Author Response · Authors · 2019-11-08
**General Response**

We thank all the reviewers for the thoughtful comments. We answer each reviewer’s questions individually and we have updated the draft according to the feedback.

---

### Decision · Program_Chairs · 2019-12-19

**Decision:**

Accept (Spotlight)

**Comment:**

The paper addresses individual fairness scenario (treating similar users similarly) and proposes a new definition of algorithmic fairness that is based on the idea of robustness, i.e. by perturbing the inputs (while keeping them close with respect to the distance function), the loss of the model cannot be significantly increased.
All reviewers and AC agree that this work is clearly of interest to ICLR, however the reviewers have noted the following potential weaknesses: (1) presentation clarity -- see R3’s detailed suggestions e.g. comparison to Dwork et al, see R2’s comments on how to improve, (2) empirical evaluations -- see R1’s question about using more complex models, see R3’s question on the usefulness of the word embeddings.
Pleased to report that based on the author respond with extra experiments and explanations, R3 has raised the score to weak accept. All reviewers and AC agree that the most crucial concerns have been addressed in the rebuttal, and the paper could be accepted - congratulations to the authors! The authors are strongly urged to improve presentation clarity and to include the supporting empirical evidence when preparing the final revision.